# Adaptive Grasp Pose Optimization for Robotic Arms Using Low-Cost Depth Sensors in Complex Environments

**DOI:** 10.3390/s25030909

**Published:** 2025-02-03

**Authors:** Aiguo Chen, Xuanfeng Li, Kerui Cen, Chitin Hon

**Affiliations:** 1Faculty of Innovation Engineering, Macau University of Science and Technology, Taipa, Macau SAR 999078, China; holden_sh@126.com (A.C.); 3240005919@student.must.edu.mo (X.L.); 3240007122@student.must.edu.mo (K.C.); 2The Institute of Systems Engineering, Macau University of Science and Technology, Macau SAR 999078, China

**Keywords:** grasp pose estimation, ellipsoidal modeling, robotic arm systems, nonlinear optimization

## Abstract

This paper presents an efficient grasp pose estimation algorithm for robotic arm systems with a two-finger parallel gripper and a consumer-grade depth camera. Unlike traditional deep learning methods, which suffer from high data dependency and inefficiency with low-precision point clouds, the proposed approach uses ellipsoidal modeling to overcome these issues. The algorithm segments the target and then applies a three-stage optimization to refine the grasping path. Initial estimation fits an ellipsoid to determine principal axes, followed by nonlinear optimization for a six-degree-of-freedom grasp pose. Validation through simulations and experiments showed a target grasp success rate (TGSR) of over 83% under low noise, with only a 4.9% drop under high noise—representing a 68.0% and a 42.4% improvement over GPD and PointNetGPD, respectively. In real-world tests, success rates ranged from 95 to 100%, and the computational efficiency was improved by 56.3% compared to deep learning methods, proving its practicality for real-time applications. These results demonstrate stable and reliable grasping performance, even in noisy environments and with low-cost sensors.

## 1. Introduction

Dexterous manipulation of robots has broad applications in industries such as manufacturing, logistics, and healthcare. In industrial production, robots improve efficiency by performing tasks like sorting, packaging, and painting. In logistics systems, mobile robots and arms collaborate for intelligent de-stacking, picking, and sorting. In daily life, service robots assist with tasks such as serving tea or organizing, offering significant convenience.

Industrial robots based on artificial intelligence face significant challenges in efficiently grasping diverse objects [1,2,3], and while current grasping techniques are effective in specific scenarios, they often fail to meet the varied demands of industrial applications [4,5,6]. In both industrial and consumer-grade robot grasping tasks, the uncertainty in task execution can arise from several factors.

From a perception perspective, the grasping pose is derived from modeling and analyzing the scene data acquired by sensors. Whether using 2D or 3D data, strategies such as denoising, multi-frame accumulation, or improving sensor accuracy to reduce algorithmic strain may be employed to increase the success rate of task execution. However, these strategies often come with increased time, computational power, or equipment costs. To reduce these costs and make robotic arms more widely applicable, manufacturers have introduced miniaturized, low-cost 3D sensors, such as Intel’s RealSense and Microsoft’s Kinect. However, the perception accuracy of these devices is constrained by the cost, achieving only centimeter-level (5–10 cm) measurement precision. This has facilitated the widespread adoption of low-cost robots; however, the limited perception accuracy has resulted in a lack of robust algorithms for object grasping with these sensors. In fact, the trade-off between device cost, computational cost, and task execution robustness has been a central theme in the development of the robotics community. This paper focuses on the uncertainty in robotic arm grasping tasks caused by imprecise and insufficient perception, and aims to develop a cost-effective, efficient, and robust grasping algorithm, which is key to broader applications [7,8].

In terms of algorithmic principles, grasping solutions can be categorized into model-based and model-free approaches [9]. The former involves high maintenance costs [10,11], while the latter is more flexible but faces challenges in generalization [12,13]. Parallel computing has facilitated the application of deep learning in grasp detection [14], where the use of RGB-D sensors in conjunction with neural networks such as CNN and PointNet has enhanced the accuracy of grasp prediction [15,16,17,18]. However, methods like GPD and PointNet face high data dependency and computational demands, limiting their industrial applications [19,20,21,22,23]. Current methods achieve success rates of 75–95% under controlled conditions; however, this rate is insufficient for dealing with clutter and occlusion in real-world scenarios [7,19,24,25,26]. Improving the robustness and adaptability of algorithms remains a key challenge [27,28,29].

This paper proposes a robust, efficient grasp pose estimation algorithm for a robotic arm with a two-finger gripper and consumer-grade depth camera. Key contributions include the following:A PCA-based point cloud processing method for diverse target types and orientations, offering superior generalization.A grasp strategy considering both target pose and environment for successful, collision-free grasps in complex scenes.Millisecond-level grasp estimation using low-cost depth sensors, ensuring deployability with minimal size, weight, and power (SwaP) requirements.

## 2. Methods

### 2.1. System Overview

The hardware and algorithm framework are shown in Figure 1. The system uses an eye-in-hand setup with a RealSense D455 (RealSense D455 camera, Intel Corporation, Santa Clara, CA, USA 2020.) camera on an Elfin5 (Elfin5, JAKA Robotics, Shenzhen, China, 2021.) robotic arm, effective within a 0.5–2 m range for small to medium grasping tasks.

Grasping begins with the accurate localization of the target using MobileSAM for pixel-wise segmentation, followed by point cloud alignment. The system then separates the environment and target point clouds. A multi-objective optimization approach estimates the optimal grasp pose by maximizing grasp success while avoiding collisions, as further explained in Section 2.2 and Section 2.3.

### 2.2. Data Acquisition and Preprocessing

In this study, we adopt MobileSAM [30], a lightweight image segmentation network optimized for mobile devices. MobileSAM uses a convolutional neural network to extract target regions from images and integrates information across different scales to accurately segment the targets. Compared to traditional image segmentation methods, this approach significantly reduces computational costs while maintaining accuracy, making it suitable for resource-constrained devices. The model processes normalized and resized images to produce segmentation results, followed by post-processing to generate the final mask. Test results are shown in Figure 2.

Pixel alignment between the RGB camera and the depth camera is a critical step in achieving multimodal data fusion. Camera calibration provides the intrinsic and extrinsic matrices for both the RGB and depth cameras. Specifically, a calibration board is used to calibrate the two cameras, obtaining the intrinsic matrices $K_{\mathrm{RGB}}$ and $K_{\mathrm{Depth}}$ for each camera, as well as the extrinsic matrices *R* and *T* between them. The pixel coordinates $\left(u_{d},\nu_{d}\right)$ from the depth image are mapped to 3D space coordinates $\left(\dot{X},Y,Z\right)$, which are then transformed into the RGB camera coordinate system using the extrinsic matrices. Finally, these 3D coordinates are projected back into the RGB pixel coordinate system $\left(u_{r},\nu_{r}\right)$ using the intrinsic matrix of the RGB camera. This mapping relationship can be expressed by the following equations:(1)$$\left[\begin{array}{c} u_{r}\\ \nu_{r}\\ 1 \end{array}\right]=K_{RGB}\cdot \left(R\cdot \left[\begin{array}{c} X\\ Y\\ Z \end{array}\right]+T\right)$$

Through the aforementioned process, the depth image can be reprojected into the coordinate system of the RGB image, achieving pixel-level alignment. Point cloud processing is a crucial step in analyzing and manipulating 3D point data obtained from depth images or 3D scanning devices. The first step involves generating a 3D point cloud using camera calibration parameters and the depth image. Specifically, the 3D coordinates (X,Y,Z) in the camera coordinate system are calculated for each pixel (u,ν) in the depth image along with its corresponding depth value *d*. This process can be achieved using the following equations:(2)$$X=\left(u-u_{0}\right)\cdot d/f_{x}$$(3)$$Y=\left(v-v_{0}\right)\cdot d/f_{y}$$(4)Z=d
where $f_{x}$ and $f_{y}$ are the focal lengths of the camera, and $\left(u_{0},\nu_{0}\right)$ are the coordinates of the principal point.

After the above process, we can align the results from 2D image recognition with the 3D sensor data on a pixel-by-pixel basis to obtain the 3D information of the target object. Even in cases where the target overlaps with the background, the object can still be separated from the environment. However, in more complex scenarios, relying solely on instance segmentation from 2D images may not be robust enough. For example, when the object and background objects have similar colors or textures, and there is significant foreground-background overlap, the instance segmentation results on the image may suffer from boundary overflow, which could affect the extraction of the 3D information of the target. To address this, we employ a clustering strategy to exclude point clouds that do not belong to the target. The implementation principle is described below: We introduce the Euclidean clustering method, which effectively removes misidentified objects in the background under overlapping environments by considering the spatial distance information of objects in the point cloud. Even if instance segmentation confuses background objects with foreground objects, their positions in three-dimensional space typically differ significantly. Euclidean clustering calculates the Euclidean distance between points and groups closely located points into the same cluster, thereby effectively separating foreground and background objects. For each point, Euclidean clustering computes its neighborhood and assigns these points to the same cluster based on a preset distance threshold ϵ. If a background object is spatially far from a foreground object, even if it is mistakenly segmented as foreground in the 2D image, Euclidean clustering can exclude it from the foreground. After clustering, background objects, due to their greater distance from foreground objects, are typically assigned to a separate cluster or labeled as noise, thereby being discarded. Ultimately, only the point cloud close to the foreground is retained, ensuring the effective removal of background objects.

It is important to note that if the target we wish to extract is largely occluded by an unrelated object and becomes invisible, this will first pose a challenge to instance segmentation. Thus, this strategy is primarily designed for optimizing the segmentation of overlapping foreground objects. Regarding overlapping background objects, we may need to consider how to circumvent and grasp targets in a more reasonable posture, which will be discussed in later in Section 2.3.

### 2.3. Grasp Pose Estimation

This section outlines the method for estimating the grasp pose, as introduced in Section 2.2. The coordinate system of the robotic arm is illustrated in Figure 3. The prismatic axis defines the gripper’s movement, the tilt axis indicates the pitch angle, and the advance axis, derived from the cross product of the prismatic and tilt axes, defines the direction of gripper extension.

#### 2.3.1. Ellipsoid Modeling

Grasp pose estimation aims to improve success by analyzing grasp factors. Objects are grasped by their thinnest part, modeled within an ellipsoid bounding box, as shown in Figure 4.

The ellipsoid model can be represented by the following general quadratic surface equation:(5)$$Ax^{2}+By^{2}+Cz^{2}+2Dxy+2Exz+2Fyz+2Gx+2Hy+2Iz+J=0$$

Let the point cloud data be denoted as $P={\left\{\left(x_{i},y_{i},z_{i}\right)\right\}}_{i=1}^{N}$. To apply the least squares method, a design matrix A and an observation vector b are constructed as follows:(6)$$A=\left[\begin{array}{ccccccccc} x_{1}^{2} & y_{1}^{2} & z_{1}^{2} & 2x_{1}y_{1} & 2x_{1}z_{1} & 2y_{1}z_{1} & 2x_{1} & 2y_{1} & 2z_{1}\\ x_{2}^{2} & y_{2}^{2} & z_{2}^{2} & 2x_{2}y_{2} & 2x_{2}z_{2} & 2y_{2}z_{2} & 2x_{2} & 2y_{2} & 2z_{2}\\ \vdots & \vdots & \vdots & \vdots & \vdots & \vdots & \vdots & \vdots & \vdots \\ x_{N}^{2} & y_{N}^{2} & z_{N}^{2} & 2x_{N}y_{N} & 2x_{N}z_{N} & 2y_{N}z_{N} & 2x_{N} & 2y_{N} & 2z_{N} \end{array}\right],\;b=\left[\begin{array}{c} -1\\ -1\\ \vdots \\ -1 \end{array}\right]$$

Solving the linear system AX=b using the least squares method yields the parameter vector $X={\left[A,B,C,D,E,F,G,H,I,J\right]}^{T}$. These parameters define the general equation of the ellipsoid. The next step is to derive the ellipsoid’s geometric parameters, such as the center position and the lengths of its axes.

By completing the square for the ellipsoid equation, the coordinates of the center point $Q_{t}=\left(x_{0},y_{0},z_{0}\right)$ can be calculated. Assuming the ellipsoid equation is in the standard form, as follows:(7)$$A{\left(x-x_{0}\right)}^{2}+B{\left(y-y_{0}\right)}^{2}+C{\left(z-z_{0}\right)}^{2}+\ldots =1$$

If the cross-product terms (i.e., the terms involving xy, xz, and yz, such as 2Dxy, 2Exz, 2Fyz) are non-zero, it indicates that the ellipsoid is rotated relative to the coordinate axes. Specifically, the non-zero cross-product terms imply that the principal axes of the ellipsoid are not aligned with the coordinate axes but are instead rotated. In this case, a rotation matrix can be derived to transform the ellipsoid into its canonical or standard form, where the axes are aligned with the coordinate system.

The center coordinates of the ellipsoid are then given by(8)$$x_{0}=-\frac{G}{A},\;y_{0}=-\frac{H}{B},\;z_{0}=-\frac{I}{C}$$

Next, the quadratic form matrix Q is constructed as follows:(9)$$Q=\left[\begin{array}{ccc} A & D & E\\ D & B & F\\ E & F & C \end{array}\right]$$

The eigenvalue decomposition of Q is performed as follows:(10)$$Q=R\Lambda R^{T}$$
where Λ is the diagonal matrix, with its diagonal elements being the eigenvalues $\lambda_{1},\lambda_{2},\lambda_{3}$, and the square roots of the reciprocals of the eigenvalues provide the lengths of the semi-axes of the ellipsoid, denoted as a,b,c:(11)$$a=\frac{1}{\sqrt{\lambda_{1}}},\;b=\frac{1}{\sqrt{\lambda_{2}}},\;c=\frac{1}{\sqrt{\lambda_{3}}}$$
where $\lambda_{1}<\lambda_{2}<\lambda_{3}$.

The transformation matrix from the ellipsoid coordinate system to the world coordinate system, $T_{wt}$, includes both rotation and translation components:(12)$$T_{wt}=\left[\begin{array}{cc} R & t\\ 0 & 1 \end{array}\right]$$
where R is the rotation matrix composed of the eigenvectors, and *t* is the translation vector, representing the center position of the ellipsoid $\left(x_{0},y_{0},z_{0}\right)$.(13)$$\left(x_{0},y_{0},z_{0}\right)=\left(-\frac{G}{A},-\frac{H}{B},-\frac{I}{C}\right)$$

The lengths of the axes are(14)$$a=\frac{1}{\sqrt{\lambda_{1}}},\;b=\frac{1}{\sqrt{\lambda_{2}}},\;c=\frac{1}{\sqrt{\lambda_{3}}}$$

The complete transformation matrix is(15)$$T_{wt}=\left[\begin{array}{cc} R & {\left[x_{0},y_{0},z_{0}\right]}^{T}\\ 0 & 1 \end{array}\right]$$

These geometric parameters, including the rotation matrix and axis lengths, are depicted in Figure 5.

Based on the previous analysis, we can define the three-axis coordinate system of the modeled ellipsoid, with the transformation matrix from the world coordinate system to this new system being denoted as $T_{wt}$. The rotation matrix $R_{wt}=\left[R_{x},R_{y},R_{z}\right]$, where $R_{x}$ and $R_{z}$ correspond to the longest and shortest axes of the ellipsoid $\left\{C_{t}\right\}$, respectively. The initial grasp pose is configured such that the marching axis aligns with the negative direction of the *y*-axis of $\left\{C_{t}\right\}$, and the prismatic axis aligns with the *z*-axis. This configuration ensures that the gripper can effectively grasp larger objects within its limited finger opening and closing range. The tilt axis aligns with the *x*-axis, minimizing the risk of contact between the robotic arm and the target or surrounding objects. Consequently, the grasp pose transformation involves a ninety-degree rotation of the coordinate system around the *x*-axis, centered at the ellipsoid’s center, aligning the transformed *y*-axis with the original *z*-axis. The resulting coordinate vector is then multiplied by a rotation matrix representing this ninety-degree rotation about the *x*-axis. The grasp pose $T_{g}$ is subsequently defined as(16)$$T_{g}^{0}=\left[\begin{array}{cc} R_{x}\left(\pi \right) & 0\\ 0 & 1 \end{array}\right]\cdot T_{wt},R_{x}\left(\pi \right)=\left[\begin{array}{ccc} 1 & 0 & 0\\ 0 & 0 & -1\\ 0 & 1 & 0 \end{array}\right]$$

#### 2.3.2. Grasp Pose Optimization

The ellipsoid model and initial grasp pose were established, but a PCA-based model is insufficient for complex environments. Our algorithm adjusts the pitch angle based on the environment and target distance, expanding the workspace and enhancing robustness. Pitch and yaw angles are analyzed separately.

The pitch angle of the gripper is influenced by two factors: the target’s height relative to the robot. For a given grasping target, its height relative to the robot arm’s base can be represented as *h*, and the distance of the target from the robot arm’s base in the XOY plane can be expressed as $d=\sqrt{x^{2}+y^{2}}$. The parameters α and β are set based on the operational range of the robot arm. The graphical representation of this function is shown in Figure 6.

In robotic grasping systems, the pitch angle of the grasping pose critically affects the success rate and stability. To optimize grasping performance, we propose a Gaussian-based pitch angle modeling method that accounts for variations in the horizontal distance and vertical height between the target and the robotic arm’s base.

First, as the horizontal distance *d* between the target and the robot arm’s base increases, the pitch angle θ should gradually decrease, allowing the robotic arm to extend further and cover a larger workspace. Similarly, as the vertical height *h* of the target increases, the pitch angle θ should also decrease appropriately to ensure stable grasping over a wider range of heights. Based on this design concept, we model the pitch angle θ as a two-dimensional Gaussian function of the horizontal distance *d* and vertical height *h*:(17)$$\theta \left(h,d\right)=\theta_{0}\cdot \exp\left(-\frac{{\left(d-d_{0}\right)}^{2}}{2\sigma_{d}^{2}}-\frac{{\left(h-h_{0}\right)}^{2}}{2\sigma_{h}^{2}}\right)$$
where $\theta_{0}$ is the maximum value of the pitch angle, and $d_{0}$ and $h_{0}$ represent the horizontal distance and vertical height at which the pitch angle reaches its maximum value, respectively. The parameters $\sigma_{d}$ and $\sigma_{h}$ control the spread of the Gaussian distribution along the horizontal distance and vertical height, determining the rate at which the pitch angle decays as *d* and *h* change.

After obtaining the optimal pitch angle $\theta^{*}$, the grasping pose is fine-tuned, and the transformation matrix is given by(18)$$R_{x}\left(\theta^{*}\right)=\left[\begin{array}{ccc} 1 & 0 & 0\\ 0 & \cos\theta^{*} & -\sin\theta^{*}\\ 0 & \sin\theta^{*} & \cos\theta^{*} \end{array}\right]$$

With this modeling approach, the pitch angle reaches its maximum value when the target is near $d=d_{0}$ and $h=h_{0}$, and it gradually decreases following a Gaussian distribution as the target deviates in horizontal distance or vertical height. This approach integrates target position with grasp pose, maximizing workspace and enhancing flexibility. Figure 6b illustrates grasp adjustments and optimizations for different heights and distances.

#### 2.3.3. Obstacle Avoidance Design

This section builds upon the previous discussion, where we modeled the target and established the initial grasp pose, and optimized the pitch angle. Now, we focus on adjusting the yaw angle to improve obstacle avoidance in complex environments.

Figure 7 illustrates various grasping scenarios: (a) grasping a pen on a flat surface, (b) grasping a target on an inclined surface requiring yaw angle adjustment, and (c) grasping a target in a cluttered environment that demands flexible yaw optimization to avoid obstacles. This approach aims to enhance obstacle avoidance by adaptively adjusting the yaw angle based on the surrounding environment.

To optimize the grasp pose, we define the grasp pose as $T_{g}$, where the approach direction aligns with the *z*-axis, and the gripper’s opening direction is along the *x*-axis. Given the point cloud of the target object $P_{t}$ and the neighborhood point cloud $P_{r}$ within a radius *r* around the object, we denote the yaw angle of the robotic arm as α.

To assess the potential for collisions between the robotic arm and obstacles during the grasping process, we introduce an optimization objective function based on the variance of the projection distribution of the neighborhood point cloud $P_{r}$ along the *z*-axis:(19)$$J\left(\alpha \right)=\mathrm{Var}\left(\{z_{i}\left(\alpha \right)|\forall p_{i}\in P_{r}\}\right)$$

Here, $p_{i}=\left(x_{i},y_{i},z_{i}\right)$ represents any point in the neighborhood point cloud $P_{r}$, and $z_{i}\left(\alpha \right)$ denotes the projection of point $P_{i}$ on the z-axis under the yaw angle α.

To make this variance more intuitive, the objective function is rewritten as(20)$$J\left(\alpha \right)=\frac{1}{|P_{r}|}\sum_{i=1}^{|P_{r}|}{\left(z_{i}\left(\alpha \right)-\bar{z}\left(\alpha \right)\right)}^{2}$$
where $|P_{r}|$ is the number of points in the neighborhood point cloud, and $\bar{z}\left(\alpha \right)$ is the mean projection of the neighborhood point cloud on the z-axis under the yaw angle α, calculated as follows:(21)$$\bar{z}\left(\alpha \right)=\frac{1}{|P_{r}|}\sum_{i=1}^{|P_{r}|}z_{i}\left(\alpha \right)$$

By minimizing the objective function J(α), the projection of the neighborhood point cloud on the *z*-axis can be made more concentrated, thereby reducing the likelihood of collisions between the robotic arm and obstacles during the grasping process.

This optimization method not only enables the grasp pose to better adapt to the current environment but also significantly enhances the stability and success rate of the grasp in practical applications. The optimization process can be carried out using methods such as gradient descent, genetic algorithms, or simulated annealing, ultimately yielding the optimal yaw angle $\alpha^{*}$ to guide the actual grasping operation. In summary, the implementation details of our algorithm are shown in the pseudocode of Algorithm 1.
**Algorithm 1:** Grasp pose estimation and optimization.
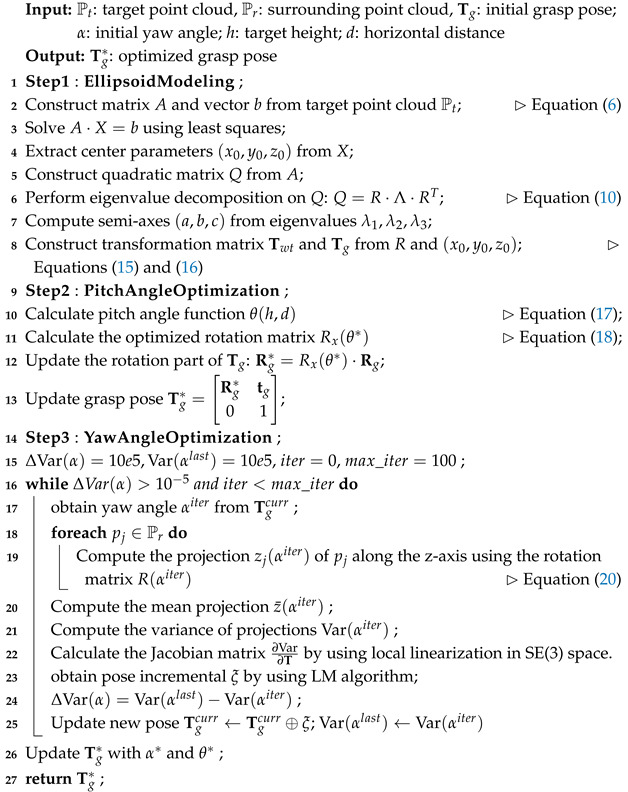


## 3. Experiment

The experiments included both simulation and real-world tests on a platform with an Intel Core i9-10700 CPU, NVIDIA GTX 2060Ti GPU, and Ubuntu 20.04.

### 3.1. Simulation Experiments

Grasp stability and success are influenced by point cloud noise and environmental complexity. Point cloud noise arises from sensor accuracy, environmental factors, and motion dynamics, which can significantly affect the performance of grasping algorithms. In real-world applications, sensor errors, variations in ambient lighting, surface reflectivity, and occlusions introduce various levels of noise, leading to inaccurate depth data or incomplete point clouds. This noise can reduce the accuracy of object recognition, pose estimation, and grasp planning, potentially resulting in grasp failures or collisions. Therefore, the robustness of the algorithm to noise is crucial for improving stability in practical scenarios.

To further assess the impact of point cloud noise on grasping performance, we conducted simulation experiments using Gazebo, a widely used platform for robotic simulation. Gazebo simulations were conducted with the robotic arm operating between 0.5 and 1.5 m, using an RGB-D camera. Gazebo, with ODE dynamics and ROS integration, provides realistic simulations and motion control.

Various common objects with different scales and shapes, including a pen, a banana, a mobile phone, an apple, a soda can, and a game controller, were selected as grasping targets. These objects were randomly placed on a table within the testing environment and tested at three different horizontal distances from the robotic arm’s origin, 0.6–0.8 m ($d_{1}$), 0.8–1.0 m ($d_{2}$), and 1.0–1.2 m ($d_{3}$), which were used as the first control variable. The experimental setup is shown in Figure 8.

To evaluate the algorithm’s stability under different point cloud noise conditions, random noise following a normal distribution was added to the coordinates of each target point, defined as $p_{i}^{\prime }=p_{i}+n_{i}$, where $P_{i}$ represents the *i*th point in the original point cloud, $p_{i}^{\prime }$ is the point with added noise, and $n_{i}∼N\left(0,\sigma^{2}\right)$ denotes Gaussian noise with a mean of 0 and a variance of $\sigma^{2}$. The noise levels were set at σ = 5 mm (low noise), σ = 10 mm (medium noise), and σ = 15 mm (high noise).

Each group was repeated 1000 times in the simulation. A successful grasp required lifting the target over 20 cm with 4 m/s² acceleration. TGSR was calculated and compared to learning-based methods GPD [19] and PointNetGPD [20]. The experimental results are shown in Figure 9, where (a), (b), and (c) represent the TGSR (%) results under point cloud noise levels of σ = 5 mm, σ = 10 mm, and σ = 15 mm, respectively. The results indicate that for medium-distance targets (0.8–1.0 m), PointNetGPD and the proposed method both achieved TGSRs above 70% in low noise, with the proposed method reaching over 83%. Under high noise, GPD and PointNetGPD rates dropped by 15.3% and 8.5%, respectively, due to reliance on point cloud stability. The proposed method showed only a 4.9% reduction, demonstrating greater robustness.

The averaged TGSR results under different noise levels are shown in Table 1. All methods performed best at medium distances, as short and long distances can cause kinematic singularities, leading to failures. The proposed method uses a nonlinear adjustment factor for adapting to different distances, achieving higher average TGSR across all scenarios.

### 3.2. Real-World Experiments

To validate the proposed method’s performance, two grasping scenarios were configured: static and dynamic. In the static scenario, an Elfin5 robotic arm with an AG_95 gripper (maximum opening width of approximately 12 cm) was used for tabletop grasping, as illustrated in Figure 10. Six objects—box, bottle, glasses case, scissors, pen, and tape—were selected as targets, representing common small to medium-sized desk items (volumes < 0.5 dm³).

The dynamic scenario, presenting greater challenges, employed a Robotiq 2F-85 gripper (maximum opening width 20 cm) mounted on a mobile platform for grasping from both the front and back of a long table. Target objects included a bottle, cup, game controller, mouse, and various fruits, representing medium-sized everyday items (volumes 0.2–1.5 dm³). Objects were densely arranged to increase the task complexity.

The Realsense-D455 was used for both experiments. The proposed method showed a 56.3% and 48.6% higher efficiency compared to GPD and PointNetGPD, relying on model optimization instead of high-performance GPUs, as shown in Figure 11.

In the static experiment, all objects were placed on a black mat, and the grasping performance of six objects was tested individually. Each target was repositioned 20 times for grasp attempts, and the success rates were evaluated. Figure 12 and Table 2 show the different grasp postures and test results for grasping a bottle, pen, and scissors.

The proposed method achieved a 100% grasp success rate for pen and glass case targets in all 20 attempts, demonstrating its stability and robustness. Other methods performed worse due to discrepancies between training and sensor-generated data, leading to incorrect grasps and poor repeatability, revealing a limitation of learning-based approaches.

In dynamic experiments, 20 targets were arranged as shown in Figure 13 and grasped sequentially, with three attempts per target. Experiment 1 grasped from the front, and Experiment 2 from the back. Success rates, calculated as the ratio of successful grasps to total attempts, are shown in Table 3.

The results presented in Table 3 indicate that the proposed method achieves a higher grasp success rate with fewer attempts, which is crucial for real-world robotic applications where precision is challenging.

## 4. Discussion

The grasp pose estimation algorithm proposed in this study represents a significant advancement in robotic arm manipulation, particularly for systems using consumer-grade RGB-D depth sensors. Extensive experiments conducted in both simulation environments and real-world scenarios demonstrate that this method significantly outperforms existing deep learning-based grasping algorithms, such as GPD and PointNetGPD, in terms of grasp success rate and computational efficiency. Specifically, under varying experimental conditions, the proposed algorithm exhibits stable performance in complex environments, while significantly enhancing computational efficiency and reducing reliance on high-performance computing resources, highlighting its broad applicability in both industrial and consumer robotics. In contrast to traditional deep learning methods, this approach does not rely on large annotated datasets or complex neural network training, resulting in lower computational costs and higher adaptability, enabling it to run on hardware platforms with limited resources.

The success of the algorithm can also be attributed to the application of nonlinear optimization techniques. By fitting ellipsoids to the point cloud, we are able to perform principal component analysis of the target’s geometric distribution, effectively describing the shape of the target object, particularly for objects with simple geometric structures. Additionally, this method incorporates the surrounding environment’s features into the nonlinear optimization for pose estimation, enabling the robotic arm to perform obstacle-avoiding grasps in the scene. However, despite its strong performance in most scenarios, the ellipsoidal model may not accurately capture the true shape of objects with complex topological structures or in dynamic environments, thereby affecting the grasp success rate. Specifically, when the surface geometry of the target object is irregular, the optimal grasp pose may not be derived solely from geometric distribution analysis, but often requires the integration of empirical constraints. For example, objects such as teapots or pots with handles may necessitate the introduction of neural network analysis at certain stages.

To enhance the stability of the algorithm in diverse environments, future research will focus on further optimizing the model to better address these challenges. Future work will concentrate on several key aspects: First, more precise geometric modeling methods, such as multimodal fusion, adaptive surface modeling, and point cloud completion, will be introduced to further improve the algorithm’s adaptability and accuracy. Second, hybrid models based on deep learning will be explored by combining traditional geometric modeling with deep learning approaches, thereby enhancing the algorithm’s generalization ability in complex scenarios.

## 5. Conclusions

In this study, the proposed grasp pose estimation algorithm demonstrated strong robustness and efficiency across multiple experiments. We simulated the impact of different sensor accuracies and environmental conditions on grasp performance by using varying levels of noise (low, medium, and high noise). Under low-noise conditions, the algorithm was able to estimate object poses with high accuracy, achieving a success rate close to 90%. As noise levels increased, the grasp success rates of other comparison methods dropped significantly, highlighting the non-negligible impact of noise on the algorithm’s performance. Under medium-noise conditions, the proposed algorithm achieved a grasp success rate (TGSR) exceeding 87%, outperforming GPD and PointNetGPD by 20% and 10%, respectively. This method effectively simplifies the modeling of object geometry, and the strategy of incorporating environmental constraints into pose optimization enables the robot to maintain high performance in cluttered environments. The algorithm also demonstrated good adaptability when handling objects of varying shapes and sizes, maintaining stable performance across different test objects.

## Figures and Tables

**Figure 1 sensors-25-00909-f001:**
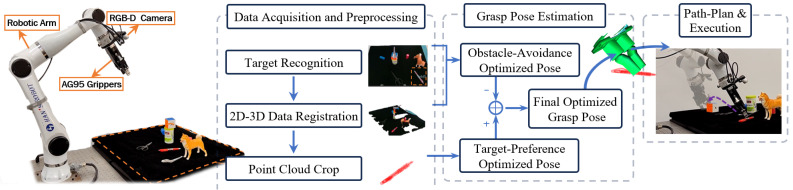
System algorithm flowchart.

**Figure 2 sensors-25-00909-f002:**
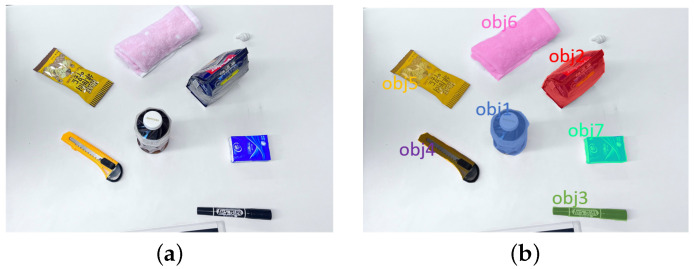
Target localization and segmentation: (**a**) original image; (**b**) segmentation result.

**Figure 3 sensors-25-00909-f003:**
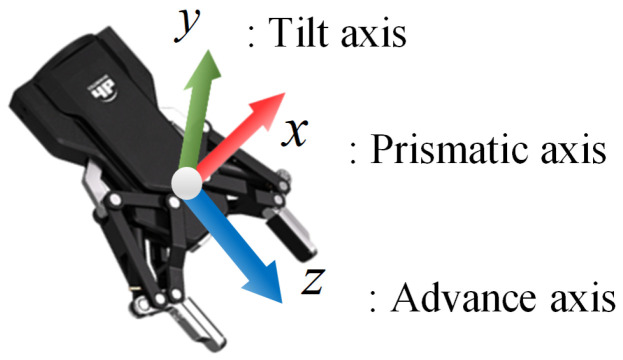
Two-finger parallel gripper coordinate system.

**Figure 4 sensors-25-00909-f004:**
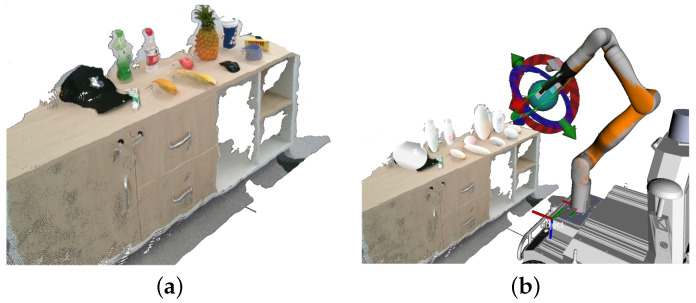
Grasping scene modeling: (**a**) point cloud of the grasping scene with registered and fused color images; (**b**) ellipsoid modeling of the grasping scene.

**Figure 5 sensors-25-00909-f005:**
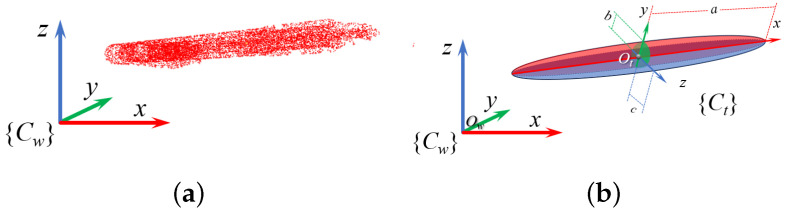
Ellipsoidal modeling of grasping target: (**a**) point cloud representation; (**b**) transformation matrix and axis lengths.

**Figure 6 sensors-25-00909-f006:**
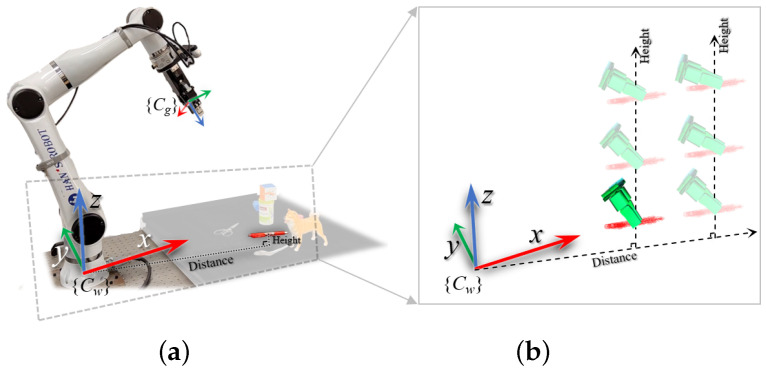
Optimization of grasp pose: (**a**) influence of target distance and height on pitch angle; (**b**) grasp adjustments for varying heights and distances.

**Figure 7 sensors-25-00909-f007:**
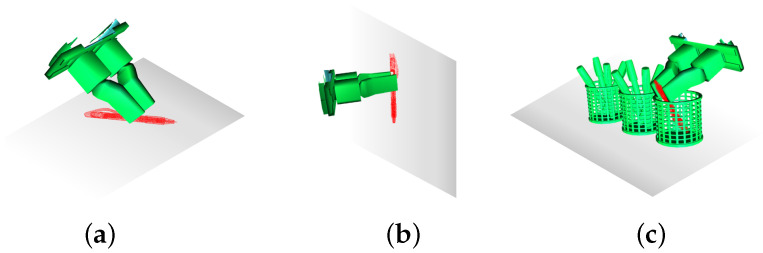
Grasp pose adjustments in complex environments: (**a**) grasping a pen on a flat surface; (**b**) grasping a target on an inclined surface; (**c**) grasping a target in a cluttered environment.

**Figure 8 sensors-25-00909-f008:**
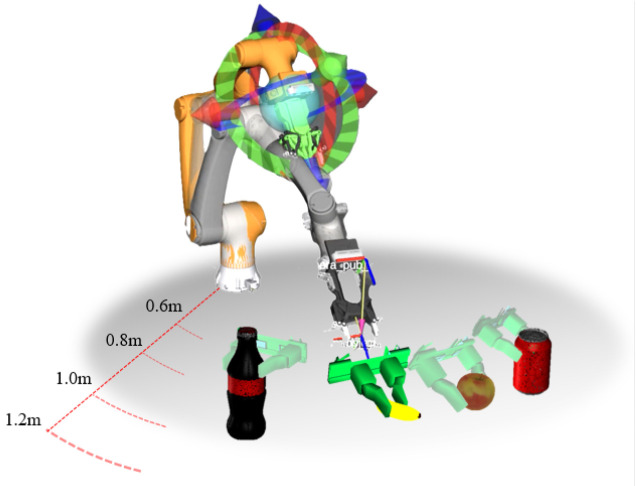
Simulation experiment setup.

**Figure 9 sensors-25-00909-f009:**
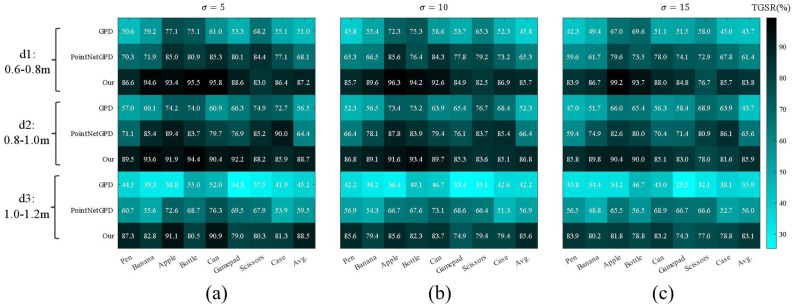
Grasp success rate (TGSR) comparison for different methods under varying noise levels: (**a**) TGSR at low noise; (**b**) TGSR at medium noise; (**c**) TGSR at high noise.

**Figure 10 sensors-25-00909-f010:**
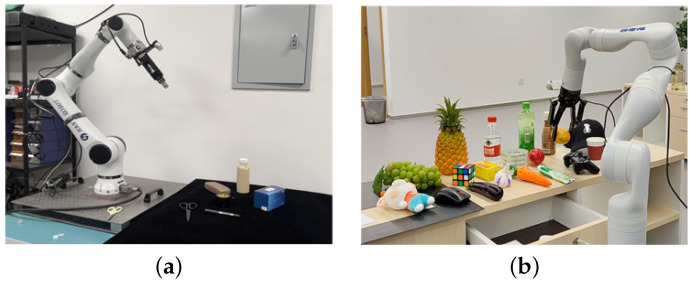
Real-world grasping scenarios: (**a**) static tabletop grasping scenario; (**b**) dynamic grasping scenario with mobile platform.

**Figure 11 sensors-25-00909-f011:**
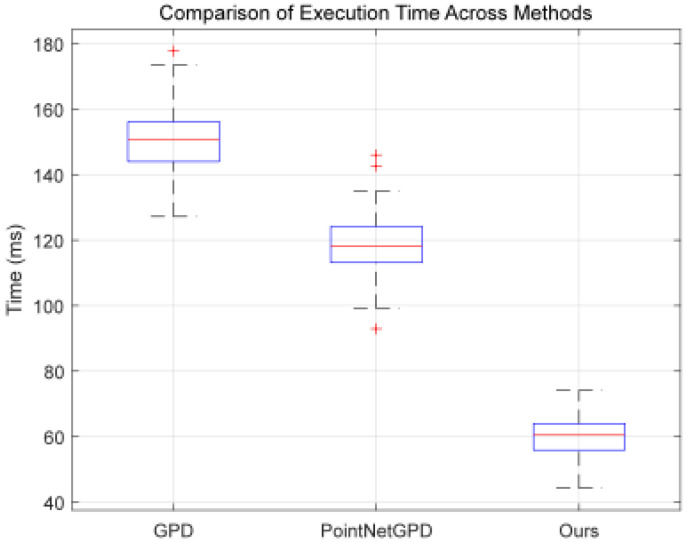
Box Plot comparing grasp pose estimation time for three methods. (The boxplot illustrates the distribution of the data, where the blue box represents the interquartile range (IQR) with the lower edge at the first quartile (Q1), the upper edge at the third quartile (Q3), and the middle line representing the median (Q2). The red line indicates the mean value of the data, while the red cross marks the outliers, which are data points significantly different from the majority of the data, typically located beyond 1.5 times the IQR from the quartiles.)

**Figure 12 sensors-25-00909-f012:**
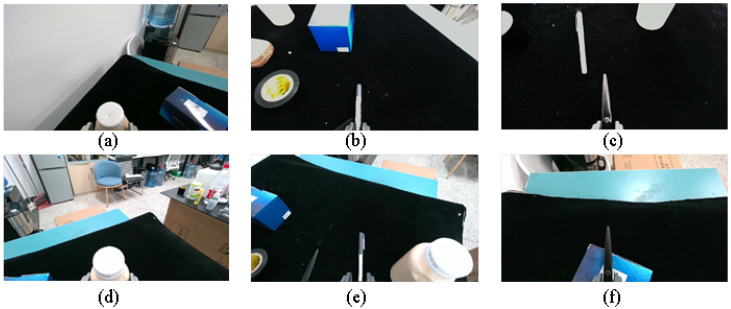
Example of target grasp execution, with images captured by Realsense D455 Color Camera, with the gripper end at the bottom of the image. (Figure (**a**): Grasping the bottle on the left side of the robot; Figure (**b**): Grasping the pen; Figure (**c**): Grasping the scissors; Figure (**d**): Grasping the bottle on the right side of the robot; Figure (**e**): Grasping the pen placed directly next to the bottle; Figure (**f**): Grasping the scissors on the box.)

**Figure 13 sensors-25-00909-f013:**
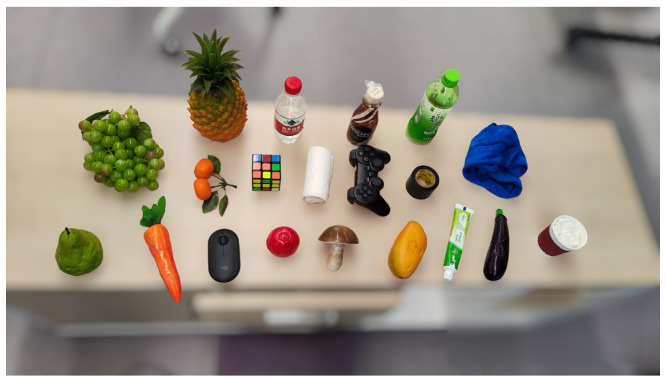
Target set in dynamic scenarios.

**Table 1 sensors-25-00909-t001:** Mean TGSR (%) in three noise environments.

Dist.	Method\Target	Pen	Banana	Apple	Bottle	Can	Gamepad	Scissors	Case	Avg.
d1	GPD	46.2	54.7	72.1	73.3	56.9	52.8	63.8	50.8	46.8
	PointNetGPD	65.1	66.7	83.4	76.9	82.6	77.3	78.8	72.7	64.9
	**Our Method**	**85.4**	**90.3**	**96.3**	**94.5**	**92.1**	**86.1**	**80.8**	**86.4**	**85.6**
d2	GPD	52.1	56.1	71.2	70.9	60.4	63.4	73.5	68.3	50.8
	PointNetGPD	65.6	79.5	86.6	82.6	76.5	74.8	83.3	87.2	65.5
	**Our Method**	**87.4**	**90.8**	**91.3**	**92.6**	**88.4**	**86.8**	**83.2**	**84.2**	**87.1**
d3	GPD	40.2	37.3	36.5	50.3	47.2	31.1	34.9	40.8	41.1
	PointNetGPD	58.0	52.9	68.3	64.3	72.8	68.3	67.0	52.6	57.5
	**Our Method**	**85.6**	**80.8**	**86.2**	**80.5**	**85.9**	**76.1**	**78.9**	**79.8**	**85.8**

**Table 2 sensors-25-00909-t002:** Target grasp success rates (%) of three methods in real-world static experiments.

Target	Box	Bottle	Glass Case	Scissors	Pen	Glue
GPD	50	45	45	50	30	55
PointNetGPD	60	65	75	55	50	75
Proposed Method	95	90	100	95	100	95

**Table 3 sensors-25-00909-t003:** Attempts count and target grasp success rates (%) of three methods in real-world dynamic experiments.

Group	GPD		PointNetGPD		Our Method	
	Attempts Count	Success Rate (%)	Attempts Count	Success Rate (%)	Attempts Count	Success Rate (%)
1	51	50	45	80	27	100
2	47	60	41	75	30	95

## Data Availability

Data underlying the results presented in this paper are not publicly available at this time but may be obtained from the authors upon reasonable request.

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
