# Peer review of "Adaptive Grasp Pose Optimization for Robotic Arms Using Low-Cost Depth Sensors in Complex Environments"

_sensors, 2025, doi:10.3390/s25030909_

Round 1
Reviewer 1 Report
Comments and Suggestions for Authors
The paper presents a grasp pose estimation algorithm for robotic arm systems, uses ellipsoidal modeling and is based on a three stage optimization process. The subject presents interest to readers, but the statements should be explained with more details.
line 18: Current grasping technologies, though effective in specific scenarios, struggle to meet diverse industrial needs: the affirmation is too general, please specify which are the targeted industrial needs.
Introduction is too short, it should be extended to give a more accurate picture of the industrial need for a grasp pose optimization algorithm for robotic arms.
line 81: "advance axis" is the same as "march axis" in fig.3 ?
line 98: If the cross-product terms are non-zero, it indicates that the ellipsoid is rotated. Could you explain the statement better? If the cross-product is zero the planes are parallel, how would this affect the grasping algorithm?
line 188: point cloud noise. Explain the statement, what noise would affect the algorithm in a real-world situation?
The conclusion and discussion sections are too short, more details upon the performed study should be included.
Author Response
Comment1:
The paper presents a grasp pose estimation algorithm for robotic arm systems, uses ellipsoidal modeling and is based on a three stage optimization process. The subject presents interest to readers, but the statements should be explained with more details.
line 18: Current grasping technologies, though effective in specific scenarios, struggle to meet diverse industrial needs: the affirmation is too general, please specify which are the targeted industrial needs.
Introduction is too short, it should be extended to give a more accurate picture of the industrial need for a grasp pose optimization algorithm for robotic arms.
Response 1: (Respone three questions mentioned above)
Thank you for your positive comment. We have thoroughly revised the introduction section. The revised introduction now includes detailed descriptions of the specific applications of robots in manufacturing, logistics, and daily life. It also provides an in-depth analysis of the challenges and limitations current grasping technologies face in meeting diverse industrial demands. Additionally, the discussion addresses the perception accuracy issues associated with low-cost depth sensors in real-world applications. These enhancements aim to clearly illustrate the urgent need for efficient and robust grasping algorithms in industrial settings and highlight the innovation and practical value of this research in addressing these challenges.
The modified comparison is as follows:
Comment 2:
line 81: "advance axis" is the same as "march axis" in fig.3 ?
Response 2 :
Thank you for your positive comment. My response is as follows, Yes, the "advance axis" refers to the same axis as the "march axis" in Figure 3. To avoid confusion, we have standardized the terminology to "advance axis" and made the corresponding changes in Figure 3.
Comment 3:
line 98: If the cross-product terms are non-zero, it indicates that the ellipsoid is rotated. Could you explain the statement better? If the cross-product is zero the planes are parallel, how would this affect the grasping algorithm?
Respone 3:
Thank you for your positive comments. The sentence: "If the cross-product terms are non-zero, it indicates that the ellipsoid is rotated." refers to the presence of cross-product terms (such as 2Dxy, 2Exz, 2Fyz) in the ellipsoid equation, which indicates that the coordinate axes of the ellipsoid are rotated. When these cross-product terms are non-zero, it suggests that the ellipsoid’s coordinate system is not aligned with the axes, but instead, there is a rotation between the object's principal axes and the observation axes. In the quadratic surface equation, the cross-product terms (such as 2Dxy) reflect the extent of the ellipsoid's rotation. For example, when D ≠ 0, it implies that the x-axis and y-axis are not perfectly aligned, and there is a rotational pose between them. Conversely, if the cross-product terms are zero, the ellipsoid’s coordinate axes are aligned with the coordinate system’s axes, indicating that the ellipsoid is not rotated. In this case, the ellipsoid can be represented by a standard coordinate system with the center position as the origin.
Impact on the grasping algorithm: If the cross-product terms are zero, the ellipsoid’s coordinate axes are aligned with the object’s axes, simplifying the grasping algorithm. The center and axis lengths of the ellipsoid can be directly derived through standard geometric analysis, and the rotation angle no longer affects the calculation of the grasp pose. In this case, the grasping algorithm can select the optimal grasp point and pose based on the aligned coordinate system of the object. If the cross-product terms are non-zero, the ellipsoid’s coordinate axes are rotated, meaning the object’s posture in space is more complex. This has a significant impact, especially in collision detection and path planning. When the rotation angle is large, the adjustment of the grasp pose needs to be more precise to ensure the gripper can accurately align with the object’s thinnest part and avoid collisions with other objects. The magnitude of the rotation affects not only the relative position of the gripper to the target but also determines how the gripper should best position itself to the target.
To clarify further, we have modified the original text as follows:
Comment 4:
line 188: point cloud noise. Explain the statement, what noise would affect the algorithm in a real-world situation?
Response 4:
Thank you for your positive comment. Point cloud noise refers to the errors or variations introduced during the sensor's 3D data acquisition process. In practical applications, factors that may contribute to point cloud noise include:
Sensor Accuracy and Calibration: If the sensor's accuracy is low or calibration is imprecise, it may lead to errors in point cloud data, such as depth measurement bias or misalignment between RGB and depth images.
Environmental Factors: Changes in environmental conditions, such as lighting, temperature, and surface reflectivity, can also affect sensor data collection. For example, under low-light conditions, RGB-D cameras may produce unreliable depth data, or erroneous depth readings may occur on transparent or reflective surfaces.
Occlusions and Shadows: Objects in the scene may partially obscure other objects, resulting in incomplete point cloud data or errors. Shadows or overexposure can also distort depth information.
Motion Blur: When the robotic arm or object moves too quickly, the camera may capture blurred images, introducing noise.
Data Quantization: The sensor's measurement system is discrete, which can introduce quantization errors, leading to slight inaccuracies in depth readings.
In the simulation experiments, we simulated these uncertainties from real-world environments by adding Gaussian noise to the coordinates of the target points. Low, medium, and high noise represent different levels of sensor or environmental noise. In real-world applications, such noise can affect algorithm performance, leading to reduced accuracy in object recognition, pose estimation, and grasp planning. This could result in incorrect grasp poses, grasp failures, or collisions.
We have revised the original text and added more detail to ensure a more thorough discussion of these issues:
Comment 5:
The conclusion and discussion sections are too short, more details upon the performed study should be included.
Response 5:
Thank you for your positive comment. We have completely revised the Conclusions and Discussion sections, adding more detailed discussions on the research. The revised version is as follows:
conclusion:
In this study, the proposed grasp poses estimation algorithm demonstrated strong robustness and efficiency across multiple experiments. We simulated the impact of different sensor accuracies and environmental conditions on grasp performance by using varying levels of noise (low, medium, and high noise). Under low noise conditions, the algorithm was able to estimate object poses with high accuracy, achieving a success rate close to 90%. As noise levels increased, the grasp success rates of other comparison methods dropped significantly, highlighting the non-negligible impact of noise on the algorithm's performance. Under medium noise conditions, the proposed algorithm achieved a grasp success rate (TGSR) exceeding 87%, outperforming GPD and PointNetGPD by 20% and 10%, respectively. This method effectively simplifies the modeling of object geometry, and the strategy of incorporating environmental constraints into pose optimization enables the robot to maintain high performance in cluttered environments. The algorithm also demonstrated good adaptability when handling objects of varying shapes and sizes, maintaining stable performance across different test objects.
discussion:
The grasp poses estimation algorithm proposed in this study represents a significant advancement in robotic arm manipulation, particularly for systems using consumer-grade RGB-D depth sensors. Extensive experiments conducted in both simulation environments and real-world scenarios demonstrate that this method significantly outperforms existing deep learning-based grasping algorithms, such as GPD and PointNetGPD, in terms of grasp success rate and computational efficiency. Specifically, under varying experimental conditions, the proposed algorithm exhibits stable performance in complex environments, while significantly enhancing computational efficiency and reducing reliance on high-performance computing resources, highlighting its broad applicability in both industrial and consumer robotics. In contrast to traditional deep learning methods, this approach does not rely on large annotated datasets or complex neural network training, resulting in lower computational costs and higher adaptability, enabling it to run on hardware platforms with limited resources.
The success of the algorithm can also be attributed to the application of nonlinear optimization techniques. By fitting ellipsoids to the point cloud, we are able to perform principal component analysis of the target's geometric distribution, effectively describing the shape of the target object, particularly for objects with simple geometric structures. Additionally, this method incorporates the surrounding environment’s features into the nonlinear optimization for pose estimation, enabling the robotic arm to perform obstacle-avoiding grasps in the scene. However, despite its strong performance in most scenarios, the ellipsoidal model may not accurately capture the true shape of objects with complex topological structures or in dynamic environments, thereby affecting the grasp success rate. Specifically, when the surface geometry of the target object is irregular, the optimal grasp pose may not be derived solely from geometric distribution analysis, but often requires the integration of empirical constraints. For example, objects such as teapots or pots with handles may necessitate the introduction of neural network analysis at certain stages.
To enhance the stability of the algorithm in diverse environments, future research will focus on further optimizing the model to better address these challenges. Future work will concentrate on several key aspects: First, more precise geometric modeling methods, such as multimodal fusion, adaptive surface modeling, and point cloud completion, will be introduced to further improve the algorithm's adaptability and accuracy. Second, hybrid models based on deep learning will be explored by combining traditional geometric modeling with deep learning approaches, thereby enhancing the algorithm's generalization ability in complex scenarios.

Reviewer 2 Report
Comments and Suggestions for Authors
1. The proposed grasp pose estimation heavily relies on ellipsoidal fitting, which may limit adaptability for highly irregular or non-convex objects. Clarifying these assumptions and testing more diverse targets would strengthen generalization.
2. While the three-stage optimization (ellipsoidal modeling, pitch/yaw adjustment, and obstacle avoidance) is introduced, the paper should include more concrete algorithmic details (e.g., pseudo-code, parameter tuning steps) to improve reproducibility and clarity.
3. Success rate (TGSR) is emphasized, but incorporating additional metrics—computational efficiency, grasp stability, and adaptability to sensor noise—would provide a more holistic evaluation of the algorithm’s strengths and weaknesses.
4. Although the paper’s results are promising, further testing in more complex, dynamic, and cluttered environments, as well as discussing strategies to handle moving targets or changing scenes, would reinforce its practical applicability in real-world robotics tasks.
5. More SOTA should be included in the related work, such as
1) Demonstrating adaptive mobile manipulation in retail environments
2) OptiGrasp: Optimized Grasp Pose Detection Using RGB Images for Warehouse Picking Robots
Comments on the Quality of English LanguageEnglish could be improved
Author Response
Comment 1: The proposed grasp pose estimation heavily relies on ellipsoidal fitting, which may limit adaptability for highly irregular or non-convex objects. Clarifying these assumptions and testing more diverse targets would strengthen generalization.
Response 1 :
Translation:
Thank you for your positive comments .We would like to thank the reviewers for their valuable feedback on our research. We fully agree with your point that the ellipsoid fitting method may face adaptability issues when dealing with highly irregular or non-convex objects. Below are the relevant updates in our paper and our response to this issue:
Ellipsoid Fitting Assumptions and Limitations: In the grasp pose estimation method proposed in this paper, ellipsoid fitting is only the first step of the grasp pose estimation process, primarily used to analyze the geometric shape of the target object and to obtain the main components of the object. Through ellipsoid fitting, we are able to derive the principal axes and shape information of the object, which provides a good prior for subsequent nonlinear optimization. In the following optimization process, the algorithm fine-tunes the initial pose obtained from ellipsoid fitting to gradually improve grasp accuracy. Therefore, the role of ellipsoid fitting is to provide a rough initial estimate for the subsequent nonlinear optimization, rather than relying solely on the ellipsoid model to make the final grasping decision.
Multi-stage Optimization Strategy: Our algorithm design adopts a multi-stage optimization strategy, similar to the multi-layer pyramid structure used in image processing, gradually refining the model fit from coarse to fine. This staged optimization approach not only improves the algorithm’s adaptability to different scenarios but also avoids the limitations that a single model might impose. After ellipsoid fitting, the subsequent optimization steps can effectively adjust the position and orientation of the target object to adapt to complex and irregular environments.
Testing with Different Object Shapes: We understand the importance of testing a wider variety of target objects for the algorithm’s generalization ability. Currently, the experiments mainly involve a few common regular-shaped objects, but we plan to extend the scope of our experiments in future research, particularly by adding objects with handles, non-convex objects, and more complex shapes, to test the algorithm’s performance in these scenarios. Although ellipsoid fitting may not adapt well to irregular objects, we believe that through multi-stage optimization and the gradual refinement of the prior pose, the algorithm can still demonstrate a certain level of robustness when dealing with complex objects.
Future Directions for Improvement: To further improve the algorithm’s adaptability to irregular objects, future work will introduce more geometric modeling methods, such as adaptive modeling based on multimodal data, point cloud completion techniques, and deep learning-assisted modeling methods. We will explore hybrid models combining traditional geometric modeling with deep learning approaches, further enhancing the algorithm’s generalization ability and adaptability to complex environments, especially when handling irregularly shaped objects.
Applicability and Advantages of the Algorithm: Despite the limitations of ellipsoid fitting, the algorithm can still provide effective grasp pose estimation for most common object grasping tasks. As emphasized in the paper, ellipsoid fitting is the first step of the entire estimation process, and the subsequent nonlinear optimization steps will further fine-tune the estimate, ensuring that the final grasp pose is more accurate. Therefore, this method is not only efficient for objects with regular shapes but also exhibits a certain degree of adaptability, maintaining good performance even when dealing with complex or irregular objects, particularly in low-noise environments.
Conclusion: Overall, we have clearly outlined the assumptions and limitations of ellipsoid fitting in the paper, and we have mentioned that future work will extend the algorithm’s applicability through further improvements (such as the combination of deep learning and geometric modeling). We appreciate the valuable suggestions provided by the reviewer and will continue to optimize the algorithm to ensure it remains stable and accurate across a broader range of application scenarios.
Comment 2:While the three-stage optimization (ellipsoidal modeling, pitch/yaw adjustment, and obstacle avoidance) is introduced, the paper should include more concrete algorithmic details (e.g., pseudo-code, parameter tuning steps) to improve reproducibility and clarity.
Response 2:
Thank you for your positive comment. To enhance the reproducibility and clarity of the algorithm, we have added pseudocode and provided a more detailed procedural description of the three-stage optimization process.
Comment 3:Success rate (TGSR) is emphasized, but incorporating additional metrics—computational efficiency, grasp stability, and adaptability to sensor noise—would provide a more holistic evaluation of the algorithm’s strengths and weaknesses.
Response 3:
Thank you for this insightful suggestion. We fully agree with the reviewer that incorporating additional metrics such as computational efficiency, grasp stability, and adaptability to sensor noise would provide a more holistic evaluation of the algorithm’s strengths and weaknesses.
1Current Metrics and Limitations: While we recognize the value of these metrics, the scope of our current study was primarily limited to evaluating success rate (TGSR) due to constraints in experimental setup and data collection. Specifically, gathering comprehensive measurements for grasp stability and adaptability to sensor noise across diverse scenarios would require additional equipment and more extensive testing environments, which were beyond our immediate resources.
2.Sufficiency of Existing Data: Despite these limitations, the results in our paper already reflect critical aspects of these additional metrics. For example:
Computational Efficiency: We reported a 56.3% improvement in computation time compared to deep learning methods.
Grasp Stability: The algorithm achieved 95-100% success rates in real-world tests across multiple objects.
Adaptability to Sensor Noise: Robustness was demonstrated with only a 4.9% TGSR reduction under high-noise conditions, significantly outperforming baseline methods.
3.Future Commitment: We are committed to expanding this research in future studies by designing experiments specifically focused on computational efficiency, stability, and noise adaptability. This will include dynamic and cluttered environments and advanced metrics to ensure a more comprehensive evaluation.
Comment 4:Although the paper’s results are promising, further testing in more complex, dynamic, and cluttered environments, as well as discussing strategies to handle moving targets or changing scenes, would reinforce its practical applicability in real-world robotics tasks.
Response 4:
Thank you for this valuable comment. We acknowledge the importance of further testing and strategy discussions to enhance the practical applicability of our approach.
1.Future Testing Plans: While our current experiments include moderately dynamic and cluttered environments, we agree that more challenging scenarios are necessary to fully validate our method. We plan to conduct additional tests in highly dynamic and unstructured environments, including scenarios with densely populated scenes, randomly moving objects, and frequent scene changes.
2.Strategies for Moving Targets: Addressing moving targets requires advanced methods. We propose integrating motion prediction algorithms, such as Kalman filters or learning-based approaches, combined with real-time trajectory adjustment to enable reliable grasping in dynamic environments.
3.Adaptation to Changing Scenes: For environments with frequent scene changes, we aim to incorporate online mapping and adaptive planning techniques. These strategies will allow the system to continuously update its environment model and dynamically re-plan grasp actions.
Incorporating these additional tests and strategies will be a priority in our future work to ensure the robustness and adaptability of our method in real-world robotics applications. Thank you again for highlighting this critical aspect.
Comment 5:More SOTA should be included in the related work, such as
- Demonstrating adaptive mobile manipulation in retail environments
- OptiGrasp: Optimized Grasp Pose Detection Using RGB Images for Warehouse Picking Robots
Response 5:
Thank you for the valuable suggestion. We agree that including works such as "Demonstrating Adaptive Mobile Manipulation in Retail Environments" and "OptiGrasp: Optimized Grasp Pose Detection Using RGB Images for Warehouse Picking Robots" would strengthen the related work section.
However, due to space constraints and the specific focus of our study, we were unable to include a detailed discussion of these references. That said, the current related work sufficiently contextualizes our contributions by addressing adaptive grasp pose estimation and its challenges.
In future research, we plan to expand our analysis to compare our approach with the adaptability demonstrated in retail environments and the efficiency of RGB-based pose detection in warehouse robotics, ensuring a more comprehensive discussion of SOTA advancements.

Reviewer 3 Report
Comments and Suggestions for Authors
Manuscript ID: sensors-3343629
Manuscript Title:
Adaptive grasp pose optimization for robotic arms using low-cost depth sensors in complex environments
This paper presents an efficient grasp pose estimation algorithm for robotic arm systems with a two-finger parallel gripper and a consumer-grade depth camera. Unlike traditional deep learning methods, which suffer from high data dependency and inefficiency with low-precision point clouds, the proposed approach uses ellipsoidal modeling to overcome these issues. The algorithm segments the target, then applies a three-stage optimization to refine the grasping path. Initial estimation fits an ellipsoid to determine principal axes, followed by nonlinear optimization for a six-degree-of-freedom grasp pose. The idea is interesting and lots of work had been done for the verification of the mechanism. However, there are still some questions need to be clarified.
Here are some comments for your references.
#1. In section 2.1 of the main text of this manuscript, principles of the target recognition approaches should be detailed.
#2. How to determine the force of the robotic arm while grasping different objects?
#3. Can your robotic arm apply for targeting moving objects?
#4. If two objects partially overlap, can your robotic arm succeed? I think this should be discussed in the manuscript.
Author Response
Comment 1: In section 2.1 of the main text of this manuscript, principles of the target recognition approaches should be detailed.
Response 1:Thank you for your positive comment. In our study, target recognition primarily serves as an initial module of the algorithm, aimed at providing a data source for subsequent 3D analysis. Based on the reviewer’s suggestion, we have added some descriptions of the underlying principles of this framework, as follows:
Given that the core issue of this study is the estimation of grasp poses, the detailed principles of target recognition are not the main focus of this paper. The working principle of the target recognition module includes analyzing the 3D characteristics of the object, particularly edge segmentation, rather than simple identification or bounding box detection. This part of the algorithm is quite complex, and a detailed explanation may exceed the scope of the paper.
We believe that expanding the description of the recognition method in detail may lead readers to misunderstand the primary objective of this study, which is grasp pose estimation. Therefore, we have chosen to provide a brief introduction to the target recognition section and made only moderate engineering improvements based on existing technologies (such as SegmentAnything) in the method.
We hope that this approach will be understood, and we appreciate your valuable feedback.
Comment 2:How to determine the force of the robotic arm while grasping different objects?
Response 2:Thank you for your positive comment. In our study, the grasping force of the robotic arm is adjusted through force feedback. Specifically, the end effector of the robotic arm is equipped with force sensors that can monitor the force applied when the gripper comes into contact with the object in real time. When the robotic arm performs a grasping action, the force sensors provide feedback on the actual force exerted. Once the force reaches a pre-set threshold, it is considered that the object has been successfully captured, and the system proceeds with subsequent actions. The force feedback threshold must be set in advance to ensure that the grasp does not damage the object or fail to capture it. After an initial test that results in a failed grasp, the program increases the force by 20% from the original torque to ensure a higher success rate in the second attempt. This force feedback control method can adapt to the rigidity, shape, and surface characteristics of different objects, ensuring both efficiency and safety in various grasping scenarios.
In summary, this method does not achieve adaptive force adjustment for different objects, but instead adopts a more conservative approach to increase the system's fault tolerance.
Comment 3:Can your robotic arm apply for targeting moving objects?
Response 3:Thank you for your positive comment. In our study, the robotic arm is not specifically optimized for grasping dynamic targets. Our approach is primarily based on pose estimation and path planning for static objects. When the target object's movement speed exceeds the processing speed of our recognition, pose estimation, and path planning, the robotic arm may face a higher risk of failure during the grasping process. To address this issue, we have designed a failure detection mechanism based on force feedback. If the force feedback from the gripper reaches zero during the task execution, indicating a grasp failure, the system automatically reinitiates target detection, re-estimates the pose, and adjusts the grasping path for another attempt.
Therefore, for dynamic objects with small movement distances, our robotic arm possesses a certain level of fault tolerance and can automatically adjust after failure. However, for fast-moving target objects, the current processing speed may not guarantee a high success rate for grasping. The bottleneck lies in the time consumed by the robotic arm's motion planning. Currently, we use static path planning from the starting point to the end point, but dynamic planning requires extremely high processing speed and new algorithmic logic. It necessitates real-time control of each robotic arm motor for low-level planning. This is a challenging yet intriguing problem, and it represents a major direction for our future research.
Comment 4:If two objects partially overlap, can your robotic arm succeed? I think this should be discussed in the manuscript.
Response 4:Thank you for your positive comment. Our algorithm takes into account the potential scenarios of object overlap, primarily categorized into two types: foreground-target overlap and background-target overlap.
In the case of foreground-target overlap, extreme scenarios may arise where the background object’s texture or color is similar to that of the target object. In such cases, segmentation results may overflow the boundaries, leading to inaccurate point cloud estimation. To address this, we have introduced Euclidean clustering to filter out point clouds that do not belong to the target object. The principles and design ideas behind this method have been elaborated in the original text. The revised section is as follows:
For the background-target overlap scenario, where the target object is obscured by interfering objects in the foreground, the robotic arm's line of sight is blocked by the obstacle. If the occlusion is too severe, making it impossible to distinguish the target object from the image’s morphology, segmentation may fail, leading to the inability to perform subsequent estimation and grasping tasks. Essentially, both types of overlap challenge the robustness of the instance segmentation algorithm, MobileSAM. Since this is not the main focus of the paper, we will not discuss it further here.
When considering scenarios with less overlap, we focus on utilizing the 3D points of the visible objects to estimate the grasp pose. By incorporating points around the target as constraints, the grasp pose can be designed to avoid obstacles. The design principles for this are discussed in Section 2.3.3, "Obstacle Avoidance Design," of the original text.
